# *Francisella* requires dynamic type VI secretion system and ClpB to deliver effectors for phagosomal escape

Maj Brodmann[1],[*], Roland F. Dreier[1],[*], Petr Broz[1] & Marek Basler[1]

*Francisella tularensis* is an intracellular pathogen that causes the fatal zoonotic disease tularaemia. Critical for its pathogenesis is the ability of the phagocytosed bacteria to escape into the cell cytosol. For this, the bacteria use a non-canonical type VI secretion system (T6SS) encoded on the *Francisella* pathogenicity island (FPI). Here we show that in *F. novicida* T6SS assembly initiates at the bacterial poles both *in vitro* and within infected macrophages. T6SS dynamics and function depends on the general purpose ClpB unfoldase, which specifically colocalizes with contracted sheaths and is required for their disassembly. T6SS assembly depends on *iglF*, *iglG*, *iglI* and *iglJ*, whereas *pdpC*, *pdpD*, *pdpE* and *anmK* are dispensable. Importantly, strains lacking *pdpC* and *pdpD* are unable to escape from phagosome, activate AIM2 inflammasome or cause disease in mice. This suggests that PdpC and PdpD are T6SS effectors involved in phagosome rupture.

[1] Focal Area Infection Biology, Biozentrum, University of Basel, Klingelbergstrasse 50/70, CH-4056 Basel, Switzerland. * These authors contributed equally to this work. Correspondence and requests for materials should be addressed to P.B. (email: petr.broz@unibas.ch) or to M.Ba. (email: marek.basler@unibas.ch).

Francisella tularensis is a Gram-negative bacterium that causes the zoonotic disease tularaemia in human and animal host. The severity of tularaemia varies depending on the route of infection and the type of strain. The *Francisella tularensis* subsp. *tularensis* is the most virulent strain and aerosol transmission of a few bacteria can cause lethal pneumonia in humans[1]. Given the low infectious dose and the severity of the infection, subsp. *tularensis* has been classified as Tier 1 select agent. The related strain *Francisella tularensis* subsp. *novicida* (*F. novicida*) has in contrast low virulence in humans, but is highly virulent in mice and thus often used as a laboratory model for tularaemia[2]. The pathogenicity of both *Francisella* species is linked to their ability to replicate in the cytosol of phagocytes, such as macrophages or dendritic cells. After phagocytosis, the bacteria shortly reside within a membrane-bound phagosome, but subsequently disrupt the phagosomal membrane and escape into the host cell cytosol, where they replicate[3].

While phagosomal escape is essential for *Francisella* intracellular replication and virulence *in vivo*, it also allows the host to mount anti-microbial and innate immune defenses. Among these are the production of type I interferons (type I IFNs) via the cGAS-STING-IRF3 axis, the production of antimicrobial guanylate-binding proteins (GBPs) and the activation of the AIM2 (absent in melanoma 2) inflammasome, which controls the release of mature IL-1β and IL-18 as well as the induction of host cell death through pyroptosis[4–11]. Interferon production and inflammasome activation require the recognition of bacterial DNA in the cytosol, and have been linked to the lysis of cytosolic *Francisella*. Mice deficient in these responses fail to control bacterial replication, resulting in a fatal disease[4–6,8,9]. *Francisella* virulence and the escape from the phagosomal compartment requires a gene cluster referred to as the *Francisella* Pathogenicity Island (FPI)[12]. Two nearly identical copies of the FPI are found in subspecies *tularensis*, *holarctica* and *mediasiatica*. The *F. novicida* genome contains only a single FPI copy[13], but features a related island called '*Francisella novicida* Island (FNI)'[14,15]. The FPI has been suggested to encode a non-canonical type VI secretion system (T6SS)[16,17], which based on gene content and phylogeny is proposed to represent a unique T6SS subtype (T6SS[ii])[18].

T6SS is a nanomachine capable of delivery of effector proteins across target cell membranes of both bacterial and eukaryotic cells and thus is often required for bacterial competition and pathogenesis[19–23]. One of the hallmarks of this system is its highly dynamic assembly that can be visualized by live-cell fluorescence microscopy[24,25]. Assembly of T6SS starts by formation of a membrane complex formed of TssJ, TssL and TssM[26]. This is followed by assembly of a baseplate complex from TssE, TssF, TssG, TssK and also VgrG, PAAR spike as well as TssA in some organisms[27–31]. Baseplate complex then initiates assembly of a long Hcp tube and TssB, TssC (or VipA, VipB) sheath wrapped around the tube[32]. Both spike and Hcp tube can associate with effectors and are delivered together into target cells upon rapid sheath contraction[33–39].

Even though the *F. novicida* sheath is structurally similar to the sheath of canonical T6SS of *V. cholerae*[40,41], it is unclear to what extent the canonical T6SS assembly mechanisms apply to *Francisella*. The reason is that *Francisella* T6SS is highly divergent and clear homologues of several core components are missing, such as TssE, TssF and TssG. In addition, many components such as TssK, VgrG, Hcp and PAAR have only low primary sequence homology to the canonical T6SS components. For example, IglG was recently shown to be structurally similar to PAAR proteins, which are required for T6SS function[15,29]. On the other hand, the FPI cluster contains many genes of unknown function, such as *iglF*, *iglI*, *iglJ*, *pdpA*, *pdpC*, *pdpE*, *pdpD* and *anmK*. PdpA, PdpC and PdpD were identified by mass-spectrometry as secreted by *Francisella* T6SS and PdpC/PdpD were proposed to be effectors required for phagosomal escape, intracellular growth and virulence[42–48]. Interestingly, the FPI cluster lacks a homologue of an unfoldase ClpV, which is present in all canonical T6SS clusters and recycles contracted sheaths[14,24,49,50]. Overall, the non-canonical gene composition suggests a unique mode of action of the *Francisella* T6SS.

Here we show that *F. novicida* T6SS sheath cycles between assembly, contraction and disassembly. Interestingly, the vast majority of T6SS sheath assemblies initiate close or at the cell pole. We show that ClpB colocalizes with contracted sheaths and is required for sheath disassembly, however, is dispensable for

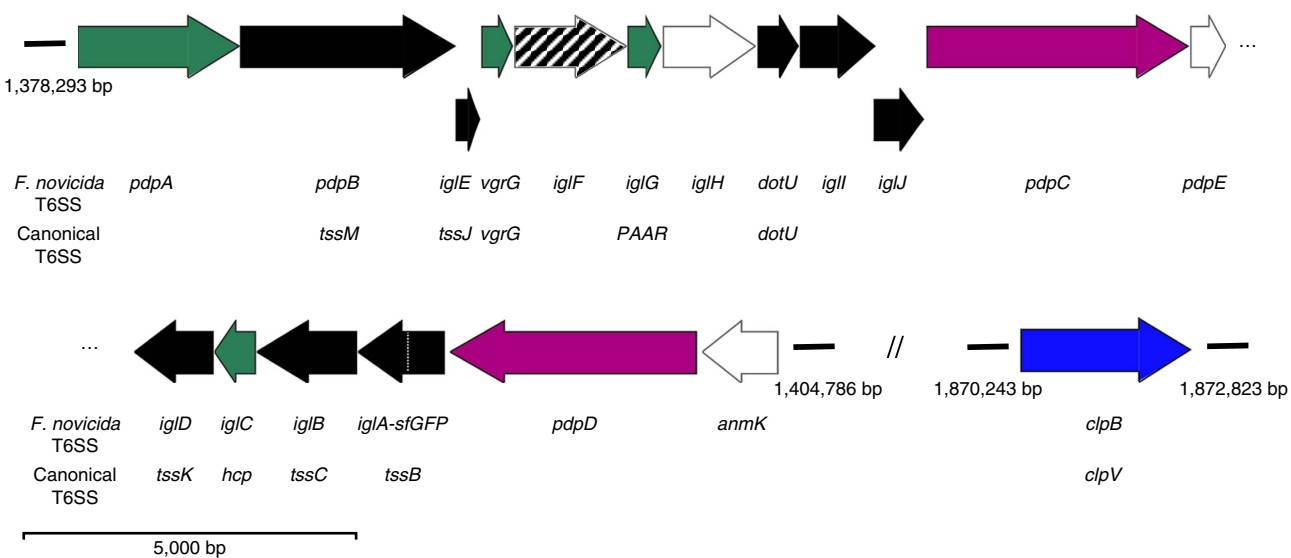

**Figure 1 | A schematic overview of *Francisella* T6SS genes.** Assignments for gene functions are based on previous studies cited in the main text and our observations: Black—structural components; Green—secreted structural components; Purple—secreted effectors; Blue—unfoldase; White—no clear evidence for function; Shaded—required for efficient assembly. The *Francisella* FPI (*pdpA–anmK*) nomenclature and the canonical T6SS nomenclature for the *F. novicida* genes is shown. Genes are drawn in scale.

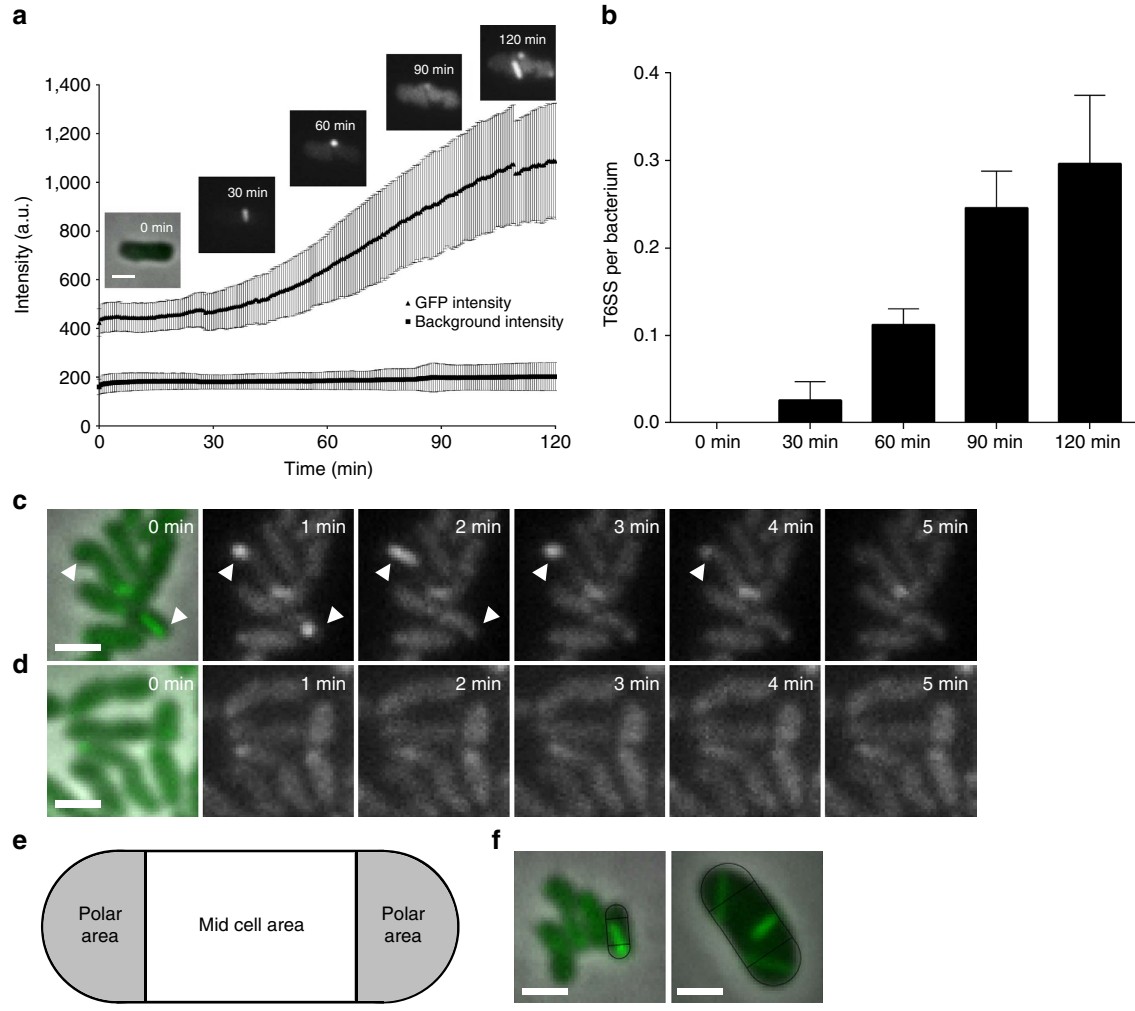

**Figure 2 | Increase of GFP intensity correlates with increased number of dynamic T6SS per bacterium.** (**a**) GFP signal intensities of *F. novicida* U112 *iglA-sfGFP* and fluorescence background were measured every minute for three regions of interest containing 1–30 bacteria. Two independent experiments were carried out. GFP intensity increase in a single *F. novicida* U112 *iglA-sfGFP* bacterium is shown at different time points. First image is a merge of phase contrast and GFP channels, following images represent GFP channel only. (**b**) Number of bacteria and T6SS structures were counted at time points between 0 and 120 min in three regions of interest containing 36–191 bacteria. Two independent experiments were carried out. Error bars represent s.d. (**c**,**d**) IglA-sfGFP localization in *F. novicida* U112 *iglA-sfGFP* wild type (**c**) and Δ*pdpB* (**d**). Arrowheads indicate T6SS sheath assembly and contraction. First image is a merge of phase contrast and GFP channels, following images represent GFP channel only. (**e**) Model for quantification of T6SS assembly position. Pole area was determined as 50% of total surface area equally distributed to both poles. (**f**) Model from **e** applied to *F. novicida* U112 *iglA-sfGFP* and *V. cholerae* 2740-80 *vipA-msfGFP*. Merge of phase contrast and GFP channels is shown. For **a**,**c**,**d** and **f** 3.3 × 3.3 μm fields of view are shown. Scale bar, 1 μm.

sheath assembly and contraction. T6SS dynamics and function depends on *iglF*, *iglG*, *iglI* and *iglJ*, while *pdpC* and *pdpD* are specifically required for phagosomal escape and virulence in a mouse model of tularaemia, but also for the engagement of the host innate immune response.

## Results

### *Francisella* T6SS is dynamic and assembles on the cell pole.
*Francisella* T6SS has a non-canonical gene composition and lacks ClpV suggesting unique mode of action (Fig. 1). To understand *Francisella* T6SS assembly and function, we searched for conditions that would allow us to image subcellular localization of TssB homologue IglA. We have serendipitously discovered that *F. novicida iglA-sfGFP* grown to an exponential phase in BHI media induced expression of IglA-sfGFP upon prolonged incubation on an agarose pad under a glass coverslip.

Importantly, the increase in expression correlated with an increase in number of IglA-sfGFP structures detected in the bacteria (Fig. 2a,b). Time-lapse imaging at a rate of 20 frames per minute showed that IglA-sfGFP structures extended across the bacteria within 30 and 120 s with assembly speeds between 5 and 15 nm s$^{-1}$. After full assembly, the IglA-sfGFP structures immediately contracted to approximately half of their original length and became brighter (Fig. 2c; Supplementary Fig. 1a; Supplementary Movies 1 and 4). After contraction, the sheath structures were disassembled during the next ∼2–3 min (Fig. 2c; Supplementary Fig. 1a). The average fluorescence intensity of the bacteria before and after one cycle of assembly, contraction and disassembly was similar, suggesting that IglA-sfGFP remained stable and folded during this cycle (Supplementary Fig. 1b). Importantly, no IglA-sfGFP structures were detected in the bacteria lacking the TssM homologue encoded by *pdpB* (Fig. 1), suggesting that assembly of IglA-sfGFP structures is dependent

on the function of the whole T6SS (Fig. 2d; Supplementary Movie 2). The dynamics of IglA-sfGFP localization is similar to that of VipA-sfGFP in *V. cholerae* and is consistent with the fact that IglA and IglB form a structure closely resembling *V. cholerae* T6SS sheath[25,40,41].

Interestingly, we also noticed that IglA-sfGFP sheaths were preferentially assembled from the bacterial pole and thus often formed structures as long as the bacterial length. To quantify the preference for subcellular localization, we divided the bacterial perimeter equally to a polar region and a mid-cell region (Fig. 2e) and counted assemblies initiated in these two equally large regions. Out of 851 assemblies, 821 assemblies (96.5%) were initiated in the polar region. As a control, we performed the same analysis for *V. cholerae* and show that only 53.8% (425 from 790) assemblies were initiated in the polar region (Fig. 2f) as expected for assemblies without preferred localization[24,25,51]. Taken together, we show that *F. novicida* assembles a dynamic T6SS sheath on the cell poles and that the sheath cycles through assembly, contraction and disassembly similarly to what was previously described for other canonical T6SSs.

**ClpB is required for disassembly of contracted sheaths.** The fact that contracted sheaths were quickly disassembled without apparent degradation of IglA-sfGFP suggested that *F. novicida* recycles contracted sheaths using a mechanism similar to the canonical ClpV-mediated sheath disassembly. The closest homologue of *V. cholerae* ClpV in *F. novicida* genome is ClpB (FTN_1743) (36% sequence identity). Interestingly, *clpB* was previously shown to be required for survival of various stresses[52] but also essential for intracellular replication and virulence of *F. novicida*[53,54].

Here we show that *F. novicida* lacking *clpB* mainly contained bright IglA-sfGFP foci (Fig. 3a). Time-lapse imaging showed that the *F. novicida* Δ*clpB* occasionally assembled new sheaths with kinetics similar to that of the parental strain but after contraction, the sheaths were never disassembled and remained intact in the bacteria (Fig. 3a,b; Supplementary Movies 1 and 4). Such assembly was still dependent on functional T6SS, as no sheath extensions and contractions were detected in *F. novicida* Δ*clpB/pdpB*. However, some bright, non-dynamic IglA-sfGFP foci were detected in the absence of both *clpB* and *pdpB* (Supplementary Fig. 2a; Supplementary Movie 2). This indicates that activity of ClpB is required for recycling of contracted sheaths, however, in case of a defect in ClpB function, some non-dynamic IglA-sfGFP foci may form also in the absence of a fully functional T6SS.

To test directly the role of ClpB in disassembly of the contracted sheaths, we introduced *clpB-mCherry2* fusion to the native locus on the chromosome of the *iglA-sfGFP* or wild-type strain. Fusing mCherry2 to ClpB had no influence on the ability of *F. novicida* to survive heat shock indicating that such fusion is fully functional (Supplementary Fig. 2d). ClpB-mCherry2 subcellular localization cycled between uniform cytosolic and punctate localization and this dynamics was dependent on the presence of *pdpB* (Supplementary Fig. 2b,c, Supplementary Movie 3). When IglA-sfGFP and ClpB-mCherry2 were imaged simultaneously, ClpB spots colocalized specifically with the contracted sheaths (Fig. 3c; Supplementary Movies 3 and 5).

*F. novicida* uses the T6SS to escape from phagosome of cells like macrophages and consistently IglA-sfGFP spots could be detected in intracellular bacteria, implying the assembly of T6SS sheaths[40]. To test whether sheath assembly is dynamic under physiological conditions during infection, we infected primary murine bone marrow-derived macrophages (BMDMs) from wild-type C57BL/6 mice for 1 h with exponentially grown

*F. novicida*. After washing away non-phagocytosed bacteria, the infected cells were fixed, stained with phalloidin and anti-*F. novicida* LPS antibody and analysed by super resolution structured illumination microscopy (SIM) to determine the relative localization of actin, bacteria and T6SS sheaths (Fig. 4a,b). This analysis confirmed that *F. novicida* reside inside the macrophage and assemble T6SS sheaths.

Next, we imaged IglA-sfGFP and ClpB-mCherry2 dynamics within *F. novicida* in live macrophages and observed that the sheaths cycled through assembly, contraction and disassembly. Importantly, ClpB-mCherry2 dynamically localized into spots that colocalized with the contracted sheaths, suggesting that ClpB is responsible for disassembly of the contracted sheaths also within phagosomes of infected macrophages (Fig. 4c; Supplementary Movie 6). In total, we analysed 30 sheath assembly, contraction and disassembly events inside live macrophages and all of the assemblies originated from the cell pole (Fig. 4c; Supplementary Movie 6). Together, these data suggest that sheath dynamics and subcellular localization observed during imaging of *F. novicida* on agarose pads is similar to that of the sheath in the bacteria residing inside of live macrophages.

To determine the importance of ClpB for *F. novicida* pathogenesis, we infected BMDMs with *F. novicida* wild-type, Δ*pdpB* and Δ*clpB* and determined the percentage of phagosomal and cytosolic bacteria using a phagosome-protection assay based on selective permeabilization of the plasma membrane with digitonin[9]. *F. novicida* Δ*clpB* had a significant defect in phagosomal escape at 4 h post infection, similarly to bacteria lacking the essential structural component PdpB (Fig. 3d; Supplementary Fig. 3a). Consistent with reduced cytosolic localization, we observed significantly reduced levels of pyroptosis induction and cytokine release in LPS-primed BMDMs infected for 10 h with *F. novicida* Δ*pdpB* and Δ*clpB*, while the wild-type strain elicited strong immune responses (Fig. 3e). Finally, we evaluated the role of ClpB *in vivo* in a mouse model of tularaemia. We infected age- and sex-matched wild-type C57BL/6 mice subcutaneously with $10^4$ colony-forming units (CFUs) of *F. novicida* wild-type, Δ*pdpB* and Δ*clpB* and measured the bacterial burden at 2 days post infection. Mice infected with *F. novicida* Δ*clpB* displayed significantly reduced bacterial counts in the liver and spleen as compared to the mice infected with *F. novicida* wild type, and in many cases no bacteria could be recovered, similarly to what was observed with *F. novicida* Δ*pdpB* (Supplementary Fig. 3b). Overall these results indicate that ClpB acts as an unfoldase for the FPI-encoded T6SS sheath, and that its activity is essential for T6SS dynamics and consequently *F. novicida* virulence.

**Differential requirement of FPI genes for sheath dynamics.** Almost all FPI genes were shown to be required for intracellular replication probably due to a lack of phagosomal escape, however, many genes of the FPI cluster have no known homologues or were not characterized in detail[14]. Importantly, both structural components of T6SS as well as putative effectors secreted by T6SS are in principle essential for overall T6SS function, however, effectors may be to a certain degree dispensable for T6SS assembly. To provide an insight into which FPI genes are required for assembly of T6SS and which may potentially encode secreted effectors, we generated in-frame deletions of genes for which we were unable to predict function based on homology to known canonical T6SS components (Fig. 1). IglA-sfGFP subcellular localization was then imaged in those strains under the same conditions as used before for the parental strain.

In Δ*iglF* and Δ*iglG* strains, we detected on average 1 dynamic sheath assembly per 400 and 500 cells, respectively, in 5 min (Supplementary Movie 2). This suggests that IglF and IglG may

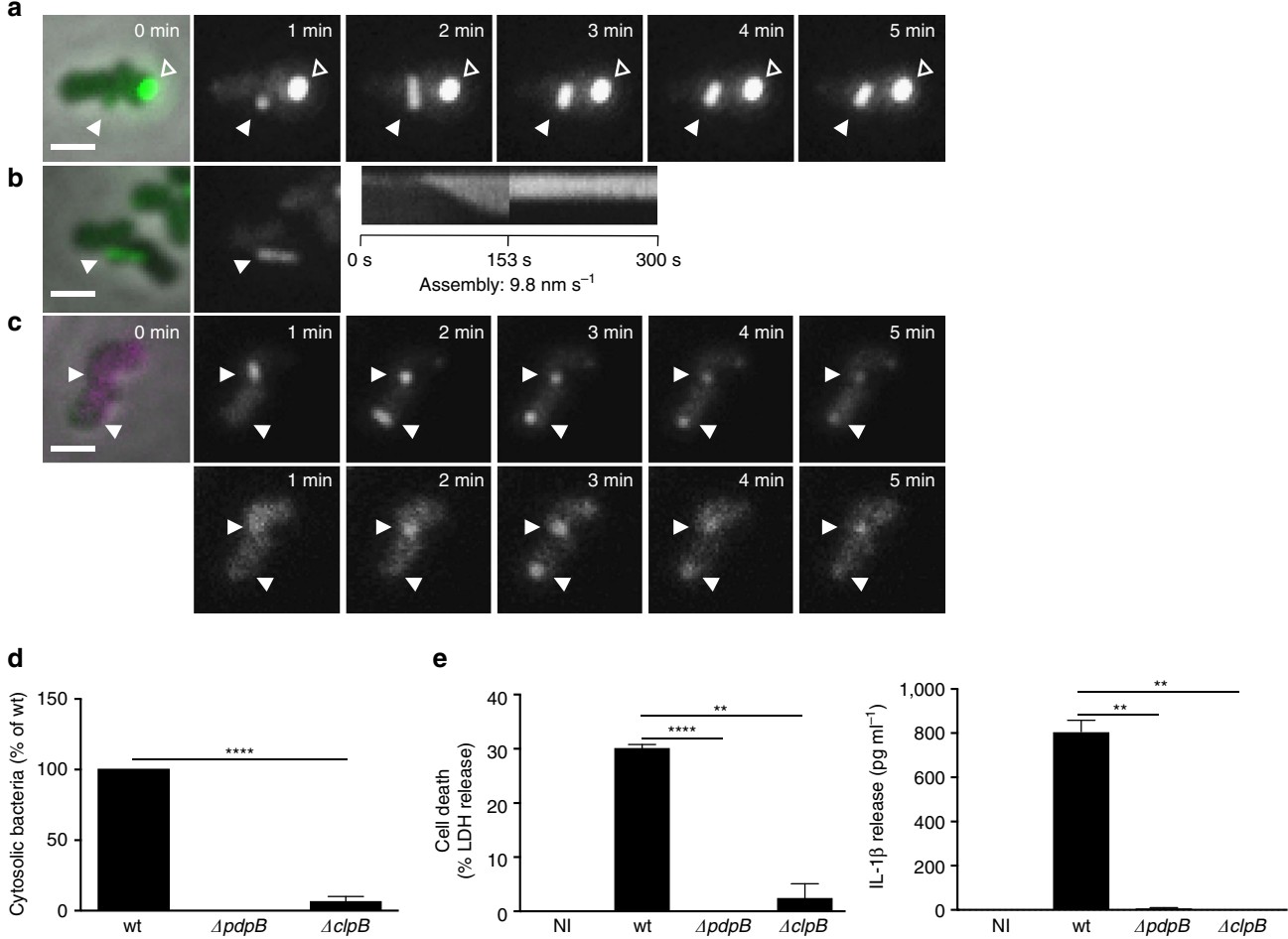

**Figure 3 | Phagosomal rupture and AIM2 inflammasome activation is dependent on disassembly of T6SS sheaths by ClpB.** (**a**) T6SS dynamics in *F. novicida* U112 *iglA-sfGFP ΔclpB*. Arrowheads indicate T6SS sheath assembly, contraction and location of sheath after contraction. Empty arrowheads indicate non-dynamic IglA-sfGFP foci. First image is a merge of phase contrast and GFP channels, following images represent GFP channel only. (**b**) Kymogram of *F. novicida* U112 *iglA-sfGFP ΔclpB* over 5 min (3 s per pixel). First image is a merge of phase contrast and GFP channels, following images represent GFP channel only. (**c**) Colocalization of ClpB-mCherry2 with IglA-sfGFP (arrows) in *F. novicida* U112 *iglA-sfGFP clpB-mCherry2*. First image is a merge of phase contrast, GFP and mCherry channels, following images represent GFP channel (upper panel) and mCherry channel (lower panel). (**d**) Quantification of cytosolic bacteria in unprimed wild-type BMDMs 4 h after infection with *F. novicida* U112 *iglA-sfGFP* wild type, *ΔpdpB* or *ΔclpB* (normalized to wild type). (**e**) Release of LDH and mature IL-1β from primed wild-type BMDMs 10 h after infection with *F. novicida* U112 *iglA-sfGFP* wild type, *ΔpdpB* or *ΔclpB* (NI—noninfected control). (**a–c**) 3.3 × 3.3 μm fields of view are shown. Scale bars, 1 μm. (**d,e**) Data are pooled from three independent experiments (**d**) (mean and s.d. are shown) or representatives of three independent experiments (**e**) (mean and s.d. of triplicate wells are shown). \*\**P* < 0.01 and \*\*\*\**P* < 0.0001 (two-tailed unpaired *t*-test with Welch's correction).

be required for efficient initiation of T6SS assembly. On the other hand, *iglI* and *iglJ* are essential for sheath assembly as no sheath assemblies were detected in more than 1,000 cells in 5 min even though IglA-sfGFP was expressed to the same level as in the parental strain (Fig. 5a; Supplementary Movie 2). Consistent with the defect in T6SS assembly, we found that Δ*iglF*, Δ*iglG*, Δ*iglI* and Δ*iglJ* strains were unable to escape into the cytosol of the infected macrophages, and consequently failed to activate cytosolic innate immune signalling (Fig. 5b,c). We cannot completely rule out the possibility that the observed phenotypes of mutants are due to polar effects on expression of other T6SS genes. However, defect in intracellular growth was previously successfully complemented for *iglF*, *iglG* and *iglI* genes[55].

Single deletion of *pdpE*, *pdpC*, *pdpD* and *anmK* or deletion of both *pdpD* and *anmK* (Δ*pdpD*/*anmK*) or *pdpC* and *pdpD* (Δ*pdpC*/*pdpD*) had no significant influence on sheath dynamics or localization (Fig. 6a; Supplementary Movie 1). Only deletion of all three genes *pdpC*, *pdpD* and *anmK* in the same strain decreased

frequency of sheath assembly by 30% from an average of one structure per three cells to about one structure per five cells (Supplementary Fig. 4a). Nevertheless, sheath assemblies in Δ*pdpC*/*pdpD*/*anmK* still preferentially localized to the cell pole, assembled with a similar speed and cycled through extension, contraction and disassembly like in the parental strain (Supplementary Fig. 4b,c; Supplementary Movie 1). Importantly, Δ*pdpE* and Δ*pdpC*/*pdpD*/*anmK* assembled sheaths with dynamics undistinguishable from the parental strain within infected macrophages (Supplementary Fig. 5e,f). In conclusion, our analysis allowed us to identify FPI genes (*iglF*, *iglG*, *iglI* and *iglJ*) essential for T6SS assembly and a distinct set of FPI genes (*pdpE*, *pdpC*, *pdpD* and *anmK*) that are dispensable for T6SS assembly.

**PdpC and PdpD are required for phagosomal escape.** To test whether *pdpE*, *pdpC*, *pdpD* and *anmK* genes are required for the escape of *F. novicida* from phagosome, we infected BMDMs with

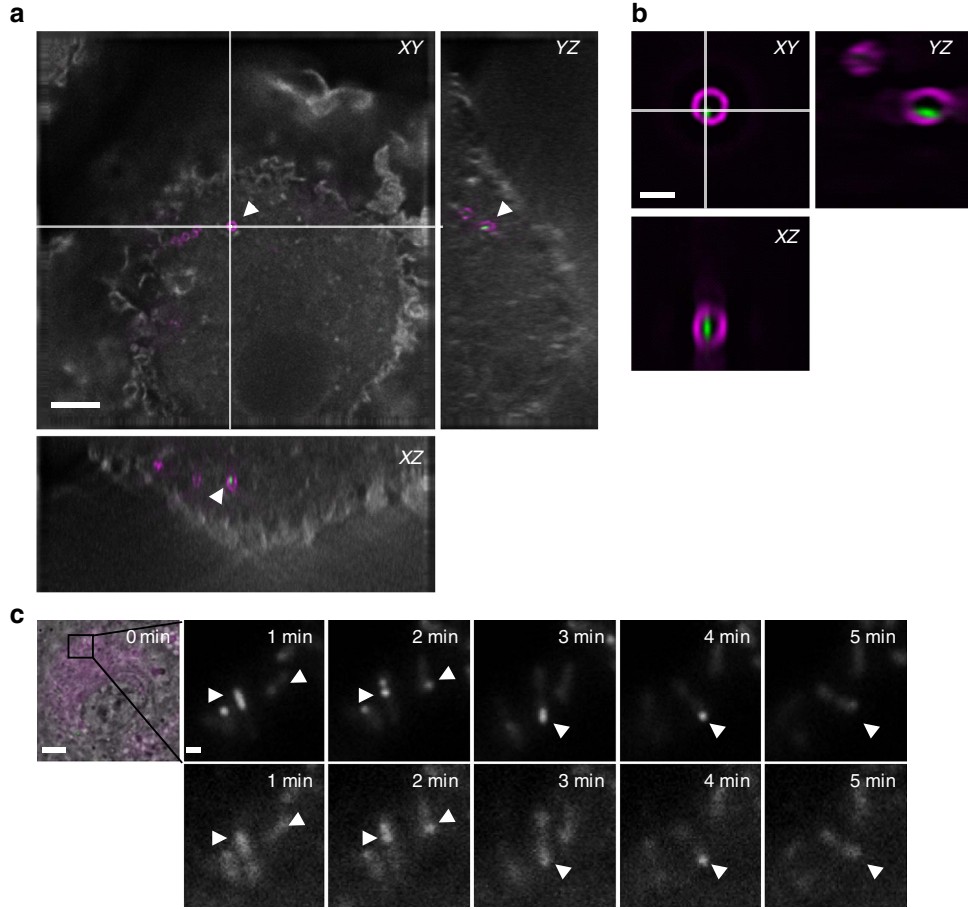

**Figure 4 | T6SS dynamics in bone marrow-derived macrophages (BMDMs).** (**a**) Merged wide field image and orthogonal view of BMDMs infected for 1 h with *F. novicida* U112 *iglA-sfGFP clpB-mCherry2*; in grey: actin staining, in magenta: LPS staining, in green: IglA-sfGFP. 41 × 41 μm field of view, scale bar, 5 μm. (**b**) Close up and orthogonal SIM view of bacterium highlighted with arrowheads in **a**; magenta: LPS staining, green: IglA-sfGFP. 5.1 × 5.1 μm field of view, scale bar, 1 μm. (**c**) Time-lapse images of unprimed wild-type BMDMs infected with *F. novicida* U112 *iglA-sfGFP clpB-mCherry2* for 1 h. First image consists of merged phase contrast, GFP and mCherry channels. 30 × 30 μm field of view is shown. Scale bar, 5 μm. Close ups consist of GFP channel (upper panel) and mCherry channel (lower panel). Close ups show 5 × 5 μm. Scale bar, 1 μm. Arrowheads indicate T6SS sheath assembly, contraction and location of sheath after contraction.

*F. novicida* Δ*pdpE*, Δ*anmK*, Δ*pdpC*, Δ*pdpD*, Δ*pdpD/anmK* or Δ*pdpC/pdpD/anmK* and determined the percentage of phagosomal and cytosolic bacteria compared to wild-type and Δ*pdpB* bacteria as outlined above (Supplementary Fig. 3a). Interestingly, we found that deletion of *pdpC* resulted in a very strong defect in phagosomal escape in comparison to wild-type bacteria, although the reduction was smaller than with bacteria lacking the structural component PdpB (Fig. 6b). *F. novicida* Δ*pdpD* and Δ*pdpD/anmK* also showed a defect in phagosomal escape, which was however less severe than the phenotype of a *pdpC* or *pdpB* deletion. No significant difference in phagosomal escape was observed between Δ*pdpD* and Δ*pdpD/anmK* strains, indicating that AnmK plays no role in phagosomal escape, consistent with the finding that phagosomal escape of the Δ*anmK* strain was indistinguishable from the wild-type strain (Fig. 6b). To determine whether the effect of a *pdpC* and *pdpD* deletion was additive, we generated a strain lacking *pdpC*, *pdpD* and also *anmK*. Interestingly, bacteria lacking *pdpC/pdpD/anmK* were unable to escape from the phagosomal compartment similarly to the Δ*pdpB* strain. In contrast, deletion of *pdpE* had no significant effect on phagosomal escape (Fig. 6b).

Next, we tested the role of *pdpE*, *pdpC*, *pdpD* and *anmK* in cytosolic innate immune detection of *F. novicida*. Consistent with the reduced level of cytosolic localization, we found that

*F. novicida* Δ*pdpC* and Δ*pdpC/pdpD/anmK* induced significantly lower levels of type I IFN production in unprimed BMDMs infected for 10 h at an MOI of 100 (Supplementary Fig. 5c). The triple mutant Δ*pdpC/pdpD/anmK* had the most severe phenotype and only elicited IFN levels in the range of the Δ*pdpB* strain (Supplementary Fig. 5c).

Since type I IFNs control the activation of the AIM2 inflammasome during *F. novicida* infection[5], we examined the level of inflammasome activation in LPS-primed infected macrophages at different time points (Fig. 6c; Supplementary Fig. 5a). While infection with *F. novicida* lacking *pdpC* or *pdpD* resulted in significantly reduced levels of inflammasome activation, only the deletion of both *pdpC* and *pdpD* completely abrogated cell death induction and cytokine production in infected macrophages, which was consistent with the reduced levels of cytosolic localization and type I IFN induction in macrophages infected with mutants lacking both proteins (Fig. 6b; Supplementary Fig. 5c). Cell death induction and cytokine production in infected macrophages was unchanged between cells infected with wild-type and Δ*anmK* bacteria indicating that AnmK is not involved in modulating inflammasome activation (Fig. 6c). Consistently, cell death and cytokine production was comparable between cells infected with *F. novicida* Δ*pdpC/pdpD* and Δ*pdpC/pdpD/anmK* or *F. novicida* Δ*pdpD* and Δ*pdpD/anmK*. Importantly, the observed

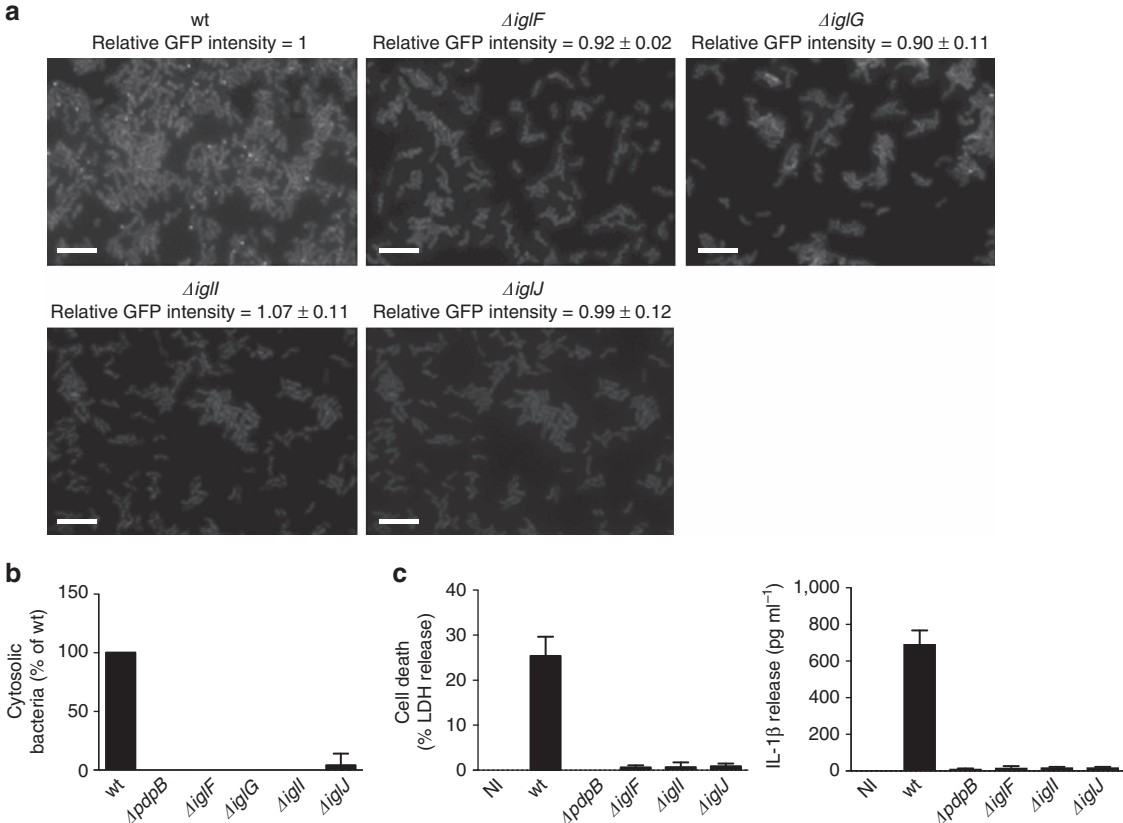

**Figure 5 | Identification of genes required for assembly and function of *F. novicida* T6SS.** (**a**) IglA-sfGFP localization in *F. novicida* U112 *iglA-sfGFP* wild type, *ΔiglF*, *ΔiglG*, *ΔiglI* and *ΔiglJ*. The GFP channel is shown. The numbers above the images represent the ratio of average GFP intensity of mutants compared to the parental strain with s.d. The average GFP intensities were quantified in three independent experiments. Thirty bacteria were analysed per experiment; 39 × 26 μm fields of view are shown. Scale bars, 5 μm. (**b**) Quantification of cytosolic bacteria in unprimed wild-type BMDMs 4 h after infection with *F. novicida* U112 *iglA-sfGFP* wild type, *ΔpdpB*, *ΔiglF*, *ΔiglG*, *ΔiglI* or *ΔiglJ* (normalized to wild type). (**c**) Release of LDH and IL-1β from primed wild-type BMDMs 10 h after infection with *F. novicida* U112 *iglA-sfGFP* wild type, *ΔpdpB*, *ΔiglF*, *ΔiglI* or *ΔiglJ* (NI—noninfected control). (**b,c**) Data are pooled from three independent experiments (**b**) (mean and s.d. are shown) or are representatives of three independent experiments (**c**) (mean and s.d. of triplicate wells are shown).

changes in inflammasome activation were independent of macrophage priming, since unprimed macrophages infected with wild-type or mutant *F. novicida* responded similarly (Supplementary Fig. 5b). Deletion of *pdpE* had no significant effect on the level of type I IFN induction, pyroptosis and cytokine release (Supplementary Fig. 5a–c).

Previous work has implicated the FPI in intracellular replication[55], therefore, we also examined intracellular replication of wild-type or mutant *F. novicida*. We monitored growth over 24 h of infection in BMDMs lacking the inflammasome adaptor protein ASC as they fail to trigger pyroptosis in response to bacterial infection[8]. *F. novicida* wild-type and *ΔpdpE* replicated over the course of the infection (Supplementary Fig. 5d), while bacteria that lacked a dynamic T6SS (*ΔpdpB* or *ΔclpB*) or bacteria that had a dynamic T6SS, but were deficient in phagosomal escape (*ΔpdpC* or *ΔpdpC/pdpD/anmK*), were cleared over the course of the infection. Consistent with reduced phagosomal escape, *pdpD/anmK*-deficient bacteria also displayed a reduced rate of replication compared to wild-type bacteria, however, the difference was not significant.

Finally, we examined the role of potential T6SS effectors *in vivo*. Age- and sex-matched wild-type C57BL/6 mice were infected subcutaneously with $10^4$ CFUs of *F. novicida* wild-type or strains deficient for the putative effectors, and the bacterial burden in the liver and spleen as well as serum IL-18 levels were assessed at 2 days post infection (Fig. 6d,e). The bacterial burden closely correlated with phagosomal escape, in that a partial reduction in virulence could be observed in *ΔpdpC* and *ΔpdpD/anmK*-infected mice. Deletion of *pdpC* alone had a stronger effect than deletion of *pdpD/anmK* although this difference was only significant in the liver. Deleting all three potential effectors, *ΔpdpC/pdpD/anmK*, rendered the bacteria largely avirulent, similarly to the deletion of the T6SS structural component *pdpB*. Consistent with the reduced levels of inflammasome activation *in vitro* (Fig. 6c; Supplementary Fig. 5a), we found that deletion of *pdpB*, *pdpC*, *pdpD/anmK* or *pdpC/pdpD/anmK* resulted in significantly lower levels of serum IL-18. A deficiency in *pdpE* appeared to have no effect on virulence or host response, since infection with *F. novicida ΔpdpE* resulted in bacterial burden and cytokine levels that were comparable to infections with *F. novicida* wild type (Fig. 6d,e). In summary, these results confirm previous studies indicating that PdpC and PdpD are T6SS-secreted effectors. Moreover, we show that PdpC and PdpD are dispensable for T6SS dynamics and specifically facilitate the escape of *F. novicida* from the phagosome into the host cell cytosol and therefore are essential for *Francisella* virulence.

## Discussion

We show here that *Francisella* T6SS sheath is under certain conditions highly dynamic and ClpB is necessary for sheath disassembly. Since ClpB-mCherry2 specifically colocalizes with

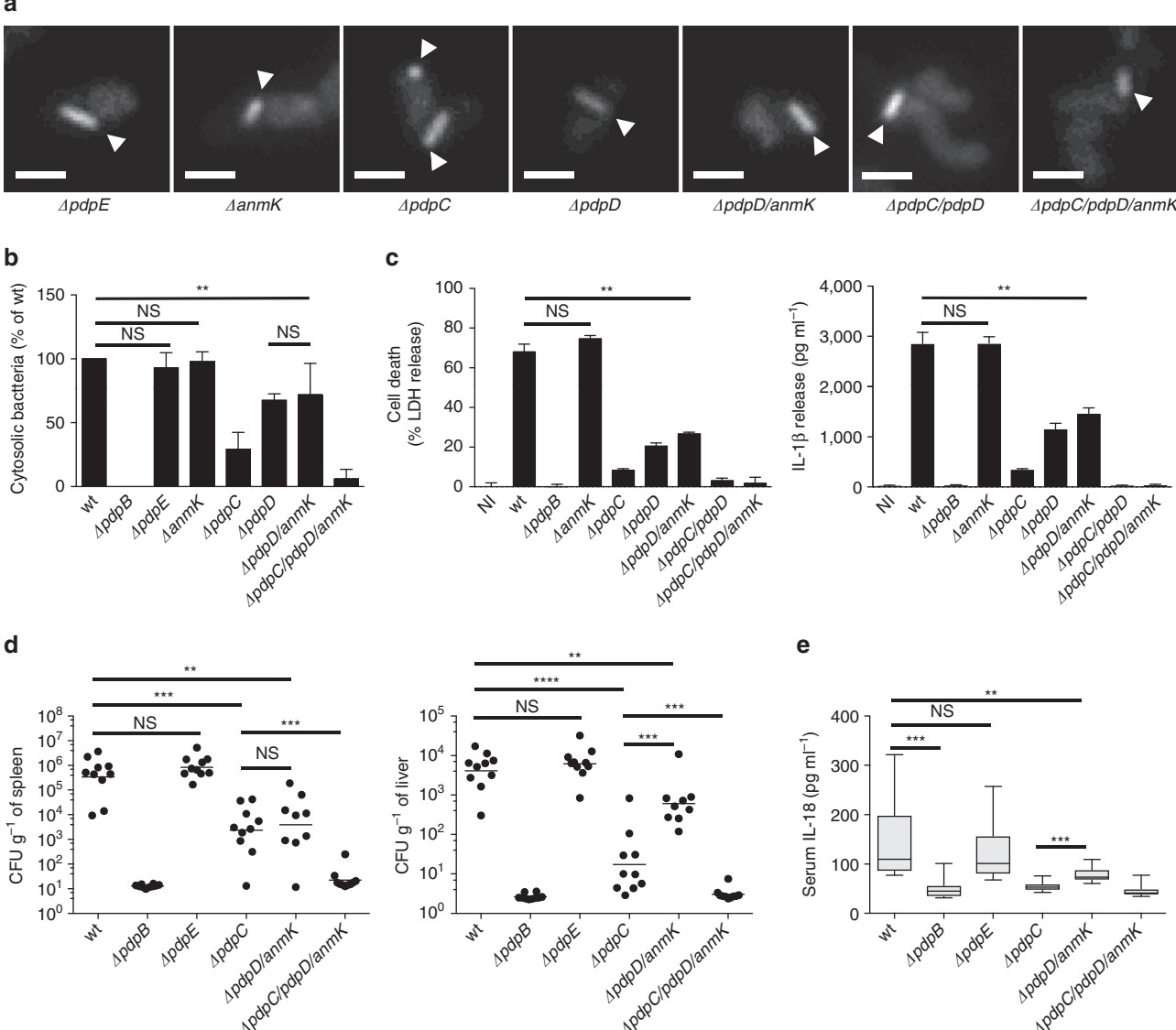

**Figure 6 | Contribution of uncharacterized FPI genes to T6SS function.** (**a**) T6SS sheath assembly (arrowheads) in *F. novicida* U112 *iglA-sfGFP ΔpdpE*, *ΔanmK*, *ΔpdpC*, *ΔpdpD*, *ΔpdpD/anmK*, *ΔpdpC/pdpD* and *ΔpdpC/pdpD/anmK*. GFP channel and 3.3 × 3.3 µm fields of view are shown. Scale bars, 1 µm. (**b**) Quantification of cytosolic bacteria in unprimed wild-type BMDMs 4 h after infection with *F. novicida* U112 *iglA-sfGFP* wild type, *ΔpdpB*, *ΔpdpE*, *ΔanmK*, *ΔpdpC*, *ΔpdpD*, *ΔpdpD/anmK* or *ΔpdpC/pdpD/anmK* (normalized to wild type). (**c**) Release of LDH and IL-1β from primed wild-type BMDMs 10 h after infection with *F. novicida* U112 *iglA-sfGFP* wild type, *ΔpdpB*, *ΔanmK*, *ΔpdpC*, *ΔpdpD*, *ΔpdpD/anmK*, *ΔpdpC/pdpD* or *ΔpdpC/pdpD/anmK* (NI—noninfected control). (**d,e**) Bacterial burden (as colony-forming units (CFUs) per gram tissue) in the spleen and liver (**d**) and serum IL-18 levels (**e**) of wild-type C57BL/6JRj mice infected subcutaneously for 2 days with 1 × 10⁴ *F. novicida* U112 *iglA-sfGFP* wild type, *ΔpdpB*, *ΔpdpE*, *ΔpdpC*, *ΔpdpD/anmK* and *ΔpdpC/pdpD/anmK*. Small horizontal lines indicate the mean. Each symbol represents an individual mouse (*n* = 5 per experiment). Graphs show pooled data from two independent biological replicates with *n* = 5 per experiment (*n* = 10 total per group); small horizontal lines indicate the mean. (**b–e**) Data are pooled from three independent experiments (**b**) (mean and s.d. are shown) or two independent experiments (**d,e**) or are representatives of three independent experiments (**c**) (mean and s.d. of triplicate wells are shown). \*\**P* < 0.01, \*\*\**P* < 0.001 and \*\*\*\**P* < 0.0001 (two-tailed unpaired *t*-test with Welch's correction (**b,c**) or Mann–Whitney test (**d,e**)).

the contracted sheaths, our data suggest that ClpB is directly involved in *Francisella* sheath disassembly similarly to ClpV in canonical T6SS (refs 24,49,50,56,57). Interestingly, *Francisella* ClpB was also shown to alter the immune response *in vivo*[58] and to be required for heat shock survival[52]. However, we show here that T6SS activity is dispensable for heat shock survival (Supplementary Fig. 2d). This suggests that, in contrast to canonical T6SS where ClpV is apparently solely dedicated to sheath disassembly, *Francisella* ClpB has a dual role. This raises the question how ClpB recognizes different substrates and

whether a specific adaptor protein is required to recognize contracted sheaths similarly to adaptor proteins that recognize substrates for AAA+-mediated unfolding[59–62]. We show that ClpB is important for *F. novicida* virulence, which is consistent with what was shown previously[52–54,58]. Since all virulence related phenotypes of *clpB*-negative strain correlated with the phenotypes of the other strains with impaired T6SS dynamics (Fig. 3; Supplementary Figs 2 and 3), we propose that *in vivo* ClpB is mainly important for T6SS sheath disassembly. However, refolding of substrates unrelated to T6SS may be required to

survive certain stresses, which *Francisella* encounter during pathogenesis.

Detailed analysis of subcellular localization of dynamic T6SS sheath shows that *Francisella* T6SS assembles on the bacterial cell poles both *in vitro* as well as during infection of macrophages (Figs 2e,f and 3c; Supplementary Fig. 5e,f). Interestingly, similarly to what we show here for *Francisella*, ClpV-5 from T6SS-5 of *Burkholderia thailandensis* was found to preferentially localize to spots on bacterial poles[63]. Those spots were however less dynamic than ClpB spots in *Francisella* and thus it remains to be directly tested if assembly of T6SS-5 of *B. thailandensis* indeed initiates at the poles. Interestingly, both T6SS-5 of *B. thailandensis* and *Francisella* T6SS are required for manipulation of the eukaryotic cells after bacterial internalization. However, unlike in *Francisella*, T6SS-5 of *B. thailandensis* is only required for formation of multinucleated giant cells after bacteria escape from endosomes using T3SS[63–67]. Since T6SS sheaths almost always assemble as long as the bacteria, one possible advantage of the polar localization could be that the sheaths assembled from the pole in rod shaped bacteria would be generally longer than the sheaths assembled from the side of the cells. Given that the T6SS sheaths only contract to about half of their extended size[25], longer sheaths may increase the distance to which T6SS can deliver effectors. Interestingly, restricted subcellular localization was shown to decrease T6SS efficiency in inter-bacterial competitions despite increased overall activity[51]. However, since *F. novicida* is completely surrounded by phagosome membrane, restricted directionality of T6SS assembly should have no consequences for delivery of effectors to the host cell. In addition, polar localization of T6SS may increase chances of puncturing phagosomal membranes, as those may be physically closer to the bacterial poles when bacteria are in a tight membrane compartment. As it was shown previously for inter-bacterial interactions, proper aiming of the T6SS apparatus at the target bacteria increases efficiency of substrate translocation[37,51].

The primary function of the *Francisella* T6SS is to promote the escape of *Francisella* from the phagosome. We show that phagosome escape depends entirely on PdpC and PdpD, which are dispensable for T6SS assembly and dynamics (Fig. 6a,b; Supplementary Movie 1), suggesting that these proteins function as effectors necessary for phagosomal escape. It is also possible that PdpD and PdpC are required for activity or secretion of yet uncharacterized T6SS effectors to promote phagosomal escape, however previous work by Eshraghi *et al.*[42] has shown that *F. novicida* PdpC and PdpD are released by the T6SS in an *in vitro* secretion assay, supporting the hypothesis that these proteins function as secreted effectors in the target cell. Moreover, *F. tularensis* and *F. holarctica* lacking *pdpC* are unable to escape from the phagosome, induce cytotoxicity and replicate intracellularly, and they are avirulent in a mouse model of tularaemia[43,44,46–48]. These observations support our conclusions that PdpC contribute to *Francisella* virulence, independent of the *Francisella tularensis* subspecies. Whereas *pdpC* is conserved in all subspecies of *Francisella tularensis*, *pdpD* is differentially encoded[45]. Therefore, PdpD might have subspecies-specific virulence related functions.

PdpC and PdpD share no homology with known effectors or pore forming toxins, such as Listeriolysin O, type C phospholipases or phenol-soluble modulins, that allow other cytosolic bacteria (*Listeria monocytogenes*, *Shigella flexneri*, *Burkholderia thailandensis* or *Staphylococcus aureus*) to escape from the phagosome, and thus might represent a novel class of effectors with membranolytic function[68,69]. The exact mechanism of how these effectors destabilize the phagosomal membrane and if this results in the recruitment of galectin-8, a marker of ruptured vacuoles that recruits antimicrobial autophagy[70], remains to be

analysed. The *Francisella* O-antigen allows the bacteria to avoid ubiquitination and uptake into LC3-positive compartments[71], but whether *Francisella* can actively inhibit or escape autophagy by injected effectors, as reported for *Listeria* and *Shigella* is unknown[72].

PdpE and AnmK, which are dispensable for T6SS assembly and phagosomal escape (Fig. 6a,b), might be effectors whose function is required once the bacteria enter the cytosol. However, their contribution to overall bacterial replication and virulence *in vivo* is minor (Fig. 6d,e; Supplementary Fig. 5d). In addition, OpiA and OpiB, encoded outside of the FPI cluster, were recently identified as T6SS secreted proteins, however, their contribution to intracellular replication is also minimal in comparison to the effects of a *pdpC* or *pdpD* deletion[42]. It is possible that these effectors have tissue-specific functions, or that they are required for *Francisella* replication in amoeba or within arthropod hosts[73,74].

Live-cell imaging of T6SS sheath dynamics suggests that IglF, IglG, IglI and IglJ are putative structural components required for T6SS assembly in *Francisella* (Fig. 5a; Supplementary Movie 2). These proteins could be homologues of components of canonical T6SS baseplate, which are difficult to identify using homology modelling[14,75] (Fig. 1). However, it is also conceivable that some of these proteins may be secreted effectors or be required for effector secretion, because deletion of certain effectors decreases T6SS function in *V. cholerae*[37,76]. Nonetheless, our finding that the dynamics of *Francisella* T6SS is possible to image *in vitro* will help to dissect the assembly of this non-canonical T6SS and to differentiate between structural components and translocated substrates. Further analysis of the structural components will reveal principles of T6SS evolution and defining the molecular mechanisms by which *Francisella* effectors modulate host cell signalling will significantly contribute to our understanding of *Francisella* virulence.

## Methods

**Bacterial strains and growth conditions.** *Francisella tularensis* subsp. *novicida* strain U112 (hereafter *F. novicida*) and the derivative strains were grown at 37 °C with aeration in brain heart infusion (BHI) medium supplemented with 0.2% L-cysteine (Sigma) and appropriate antibiotics. Antibiotic concentrations used were 100 µg ml$^{-1}$ ampicillin (AppliChem) or 15 µg ml$^{-1}$ kanamycin (AppliChem). A detailed strain list can be found in Supplementary Table 1. For infection with *F. novicida*, BHI medium was inoculated with bacteria from BHI agar plate (supplemented with 0.2% L-cysteine (Sigma) and appropriate antibiotics) and were grown overnight at 37 °C with aeration.

**Bacterial mutagenesis.** All in-frame deletions were generated by homologous recombination using the suicide vector pDMK3 as previously described[77]. A list of plasmids, primers as well as remaining peptides encoded by deleted genes can be found in Supplementary Table 2. To obtain single colonies after recombination, bacteria were grown overnight at 37 °C on Mueller-Hinton agar (MHA) supplemented with 0.1% D-glucose (Millipore), 0.1% FCS (BioConcept), 100 µg ml$^{-1}$ ampicillin (AppliChem) and 0.1% L-cysteine (Sigma) (hereafter MHA plate). Cloning product sequences were verified and chromosomal mutations were tested by PCR using primers located outside of the replaced region. Sites of homologous recombination of the chromosomal mutations were verified by sequencing.

**Heat shock survival assay.** Heat shock survival assay was adapted from ref. 52. In brief, bacteria were grown overnight as described above, diluted 1:40 in BHI medium and grown for 3 h at 37 °C with aeration. Then bacteria were diluted 1:10 in 250 µl BHI in a 1.5 ml tube and incubated in a water bath at 50 °C for 0, 15 or 30 min. At each time point the bacteria were transferred on ice and serial dilutions were plated on MHA plates. The next day, CFUs were counted and the concentration of surviving bacteria was calculated.

**Fluorescence microscopy.** Procedures and settings to detect a fluorescence signal in *F. novicida* were employed as previously described[37,41]. All imaging was carried out at 37 °C and humidity was regulated to 95% using a T-unit (Oko-lab). The exposure time was set to 150 ms for all channels. For bacterial imaging on agarose pads, *F. novicida* strains from BHI plate were washed once with BHI, diluted

1:40 in BHI medium and grown at 37 °C with aeration for 3–4 h. Bacteria from 1 ml culture were re-suspended in 50–100 μl phosphate-buffered saline (PBS), spotted on a pad of 1% agarose in PBS, covered with a cover glass (Roth) and either imaged directly or incubated at 37 °C for 1 h before imaging. Images were collected every 3 s for T6SS assembly speed quantification and every 30 s for assessment of T6SS dynamics. For imaging of infected macrophages, BMDMs were seeded onto cover glass (VWR) in 24-well plates at a density of $1.5 \times 10^5$ cells per well and infected with *F. novicida* at a multiplicity of infection of 100 in 1 ml OptiMEM (Life Technologies) as described below. Thirty minutes post infection, the BMDMs were washed three times with OptiMEM and the cover glass was mounted on a pad of 1% agarose in PBS BMDMs facing down. Images were collected every 30 s for assessment of T6SS dynamics.

**Image analysis.** Fiji[78] was used for all image analysis and manipulations as described previously[37,51]. The 'Time Series Analyzer V3.0' plugin was used for quantification of GFP signal intensity. For comparison of GFP signal intensities of mutants and wild type, only bacteria without assembled T6SS structures were considered. For quantification of T6SS activity in different mutants from 5 min time-lapse movies the 'temporal colour code' function was used. For kymograms and T6SS assembly speed quantification the 'reslice' function was used. For determination of subcellular localization of T6SS assembly the surface area of bacteria was divided into an equally sized polar and mid cell area. The surface area was calculated based on the model of a capsule using the manually measured length and width of the bacteria (see formulas below). T6SS assemblies initiating in one of the two pole areas were considered as T6SS assemblies at pole.

$$h_{\mathrm{m}} = \mathrm{Height}_{\mathrm{measured}}$$

$$l_{\mathrm{m}} = \mathrm{Length}_{\mathrm{measured}}$$

$$r = \frac{h_{\mathrm{m}}}{2}$$

$$l_{\mathrm{Cylinder}} = l_{\mathrm{m}} - h_{\mathrm{m}}$$

$$A_{\mathrm{total}} = A_{\mathrm{Sphere}} + A_{\mathrm{Cylinder}} = 4\pi r^2 + 2\pi r l_{\mathrm{Cylinder}}$$

$$A_{0.5} = 0.5 \times A_{\mathrm{total}}$$

For determination of subcellular localization of T6SS assembly, images of *V. cholerae* 2740-80 were reanalysed from ref. 37. Contrast on compared sets of images was adjusted equally. All imaging experiments were performed with at least two biological replicates.

**Structured illumination microscopy.** BMDMs were seeded onto cover glass (VWR) in 24-well plates at a density of $1.25 \times 10^5$ cells per well and infected with *F. novicida* at a multiplicity of infection of 100 for 1 h as described below. BMDMs were washed three times with PBS and fixed for 10 min at 37 °C with 4% paraformaldehyde (Electron Microscopy Science). Cover glass was incubated with chicken anti-*F. novicida* (1:2,000; a gift from D.M. Monack, Stanford University) for 1 h at room temperature, then was washed three times with PBS, incubated with goat anti-chicken coupled to Alexa 568 (1:500; Life Technologies) and DY-647-Phalloidin (1:500; Dyomics) for another 45 min at room temperature, washed three times with PBS and was mounted on glass slides with Vectashield (Vector labs). 3D-SM was performed on a microscope system DeltaVision OMX-Blaze version 4 (Applied Precision, Issaquah, WA). Images were acquired using a Plan Apo N $60 \times 1.42$ numerical aperture oil immersion objective lens (Olympus) and four liquid-cooled sCMOS cameras (pco Edge, full frame $2,560 \times 2,160$; Photometrics). Optical z-sections were separated by 0.125 μm. The laser lines 488 and 568 were used for 3D-SIM acquisition. Exposure times were typically between 10 and 140 ms, and the power of each laser was adjusted to achieve optimal intensities of between 5,000 and 8,000 counts in a raw image of 15-bit dynamic range at the lowest laser power possible to minimize photobleaching. Phalloidin Alexa-647 was acquired using the widefield mode of the system. Raw 3D-SIM images were processed and reconstructed using the DeltaVision OMX SoftWoRx software package (Applied Precision).

**Cell culture and infection.** Primary wild-type BMDMs from C57BL/6JRj mice (Janvier) were differentiated in DMEM (Sigma) with 20% M-CSF (supernatants of L929 mouse fibroblasts), 10% v/v FCS, 10 mM HEPES, nonessential amino acids and penicillin (100 IU ml$^{-1}$)/streptomycin (100 μg ml$^{-1}$) (all BioConcept). One day before infection, BMDMs were seeded into 24- or 96-well plates (Eppendorf) at a density of $1.5 \times 10^5$ or $5 \times 10^4$ cells per well in DMEM (Sigma) with 10% M-CSF (supernatants of L929 mouse fibroblasts), 10% v/v FCS, 10 mM HEPES and nonessential amino acids (all BioConcept). Where required, BMDMs were pre-stimulated overnight with LPS (from *Escherichia coli* strain O111:B4 (InvivoGen; tlr-3pelps)). *F. novicida* were grown overnight at 37 °C with aeration as described above. The bacteria were added to the BMDMs at a multiplicity of infection of 100 or the indicated value. The plates were centrifuged for 5 min at 500*g* to ensure similar adhesion of the bacteria to the cells and were incubated for

120 min at 37 °C. Next, the medium was replaced with fresh medium containing 10 μg ml$^{-1}$ gentamicin (BioConcept) to kill extracellular bacteria, then plates were incubated at 37 °C for the indicated length of time.

**Cytokine and LDH release measurement.** IL-1β and IL-18 were measured by enzyme-linked immunosorbent assay (eBioscience). Lactate dehydrogenase (LDH) was measured with an LDH Cytotoxicity Detection Kit (Takara). To correct for spontaneous cell lysis and to normalize the values, the percentage of LDH release was calculated as follows:

$$\frac{\mathrm{LDH\ value}_{\mathrm{infected}} - \mathrm{LDH\ value}_{\mathrm{uninfected}}}{\mathrm{LDH\ value}_{\mathrm{total\ lysis}} - \mathrm{LDH\ value}_{\mathrm{uninfected}}} \times 100$$

**Phagosome protection assay.** The amount of cytoplasmic and vacuolar bacteria was measured as previously described[79]. In brief, BMDMs were seeded into 24-well plates at a density of $1.5 \times 10^5$ cells per well and *F. novicida* were grown for 4 h at 37 °C with aeration as described above. BMDMs were infected with *F. novicida* at a multiplicity of infection of 100 for 4 h as outlined above. BMDMs were washed three times with KHM buffer (110 mM potassium acetate, 20 mM Hepes, 2 mM MgCl$_2$) and incubated for 1 min with 75 μg ml$^{-1}$ digitonin (Sigma) followed by differential staining of cytoplasmic and total bacteria. Antibodies used for staining were chicken anti-*F. novicida* (1:2,000; a gift from D.M. Monack, Stanford University) and goat anti-chicken coupled to Alexa 647 (cytoplasmic bacteria) or Alexa 488 (total bacteria) (1:500; both from Life Technologies). Stained bacteria were analysed on a FACS-Canto-II. Percentage of cytosolic bacteria were normalized to wild-type *F. novicida* as follows:

$$\frac{\mathrm{FACS\ value} - \mathrm{FACS\ value}_{\Delta pdpB}}{\mathrm{FACS\ value}_{wt} - \mathrm{FACS\ value}_{\Delta pdpB}} \times 100$$

**Intracellular bacterial growth assay.** BMDMs were seeded into 24-well plates at a density of $1.5 \times 10^5$ cells per well and infected with *F. novicida* at a multiplicity of infection of 1 as described above. After 2 and 24 h of infection, the BMDMs were washed three times with PBS and lysed with 0.1% Triton-X 100 (Promega) for 10 min at 37 °C. The bacteria were stained for 10 min with chicken anti-*F. novicida* (1:2,000; a gift from D.M. Monack, Stanford University), washed once with PBS and stained for 10 min with goat anti-chicken coupled to Alexa 647 and Alexa 488 (1:500 each; both from Life Technologies). A volume of 20 μl 123count eBeads (eBioscience) was added to each sample. The samples were analysed on a FACS-Canto-II by counting the number of bacteria per 5,000 beads. The CFU ratio was calculated by dividing the number of bacteria at 24 h (output) with the number of bacteria at 2 h (input).

**Type I interferon measurement.** One day before infection, ISRE-L929 reporter cells (a gift from D.M. Monack, Stanford University) were seeded into black 96-well plates with micro-clear bottom (Greiner) at a density of $1 \times 10^5$ cells per well in DMEM (Sigma) with 10% v/v FCS and penicillin (100 IU ml$^{-1}$)/streptomycin (100 μg ml$^{-1}$) (both BioConcept). BMDMs were seeded into 96-well plates at a density of $5 \times 10^4$ cells per well and infected with *F. novicida* at a multiplicity of infection of 100 as described above. After 10 h of infection, type I IFN production was measured with the Bright-Glo Luciferase Assay System (Promega) as previously described[80].

**Animal infection.** All animal experiments were approved (licence 2535-26742, Kantonales Veterinäramt Basel-Stadt) and were performed according to local guidelines (Tierschutz-Verordnung, Basel-Stadt) and the Swiss animal protection law (Tierschutz-Gesetz). Female 10 weeks old wt C57BL/6JRj mice (Janvier) were infected subcutaneously with $10^4$ CFUs of indicated stationary-phase *F. novicida* strain in 50 μl PBS. Mice were killed 48 h post infection. Bacterial load of spleen and liver was analysed by plating the bacteria on MHA plates. The plates were incubated for 24 h at 37 °C. IL-18 levels in the blood were measured by enzyme-linked immunosorbent assay (eBioscience). No randomization or 'blinding' of researchers to sample identity was used.

**Statistical analysis.** Statistical data analysis was done using Prism 6.0h (GraphPad Software, Inc.). To evaluate the difference between two groups (T6SS per cell, T6SS assembly speed, subcellular localization of T6SS, bacterial survival, cell death, cytokine release, phagosomal escape, bacterial growth and IFN production) the unpaired two-tailed *t*-test with Welch's correction was used. Animal experiments were evaluated with a two-tailed Mann–Whitney test. *P* values are given in the figure legends.

**Data availability.** The authors declare that the data supporting the findings of this study are available within the paper and its Supplementary Information files.

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

## Acknowledgements

We thank M.A. Horwitz (UCLA) for providing the strain *F. novicida* U112 *iglA-sfGFP*, D.M. Monack (Stanford University) for the conjugation plasmid, the anti-*Francisella* primary antibody and the ISRE-L929 cells, A. Harms and C. Dehio (Biozentrum, University of Basel) for the *E. coli* conjugation strain and the Imaging Core Facility and FACS Core Facility (Biozentrum, University of Basel) for technical support. We thank P.D. Ringel for his help with the analysis of subcellular localization of T6SS assembly. The work was supported by SNSF Starting Grant BSSGI0_155778 (to M.Ba.), SNSF grant 31003A_159525 (to M.Ba.), SNSF grant PP00P3_139120/1 (to P.B.) and the University of Basel. M.Br. was supported by the Biozentrum Basel International PhD Program 'Fellowships for Excellence'.

## Author contributions

M.Br., R.F.D., P.B. and M.Ba. designed experiments, analysed and interpreted the results. M.Br. and R.F.D. generated strains and acquired all data. All authors wrote and approved the manuscript.

## Additional information

**Competing interests:** The authors declare no competing financial interests.

