## [Peer Review File · Nature Communications]

Reviewers' comments:

Reviewer #1 (Remarks to the Author):

Comments:

The manuscript by Brodmann et al is focused on the T6SS dynamics and assembly and its role in anti-eukaryotic virulence in *Francisella novicida*. The authors did a very thorough dissection on the contribution of a number of previously unknown T6SS genes to the assembly, localization, and functions of T6SS using fluorescence microscopy, super-resolution microscopy, cell culture, and animal models. Novel findings include that the *Francisella* T6SS preferentially assembles on the cell pole, requires ClpB for its dynamics, and delivers putative effectors PdpC and PdpD to escape the phagosomes.

Minor comments:

1. It could be useful to include a schematic of the non-canonical T6SS gene cluster in *Francisella*, and list corresponding known T6SS homologs with conventional nomenclature.
2. Figure 3 legend – line 727, should be (A) and not (B) referring to the arrow.
3. Should include paper by Barrigan et al 2013 “Infection with *Francisella tularensis* LVS clpB Leads to an Altered yet Protective Immune Response” as a reference when discussing the importance of ClpB for *F. novicida* pathogenesis.
4. Line 268. “confirmed that PdpC and PdpD are potential...” here confirm and potential seem to be contradictory. Current data suggest PdpC and PdpD are effectors. However, can the authors comment on the possibility that PdpC and PdpD may not be effectors per se but rather as accessory proteins required for delivery of unidentified effectors?
5. This is a very minor point but it might confuse readers a little by starting the manuscript in the summary and the text with a different *Francisella* species that is not investigated in the paper. However, I will leave this to the authors judgement.

Reviewer #2 (Remarks to the Author):

Francisella pathogenesis is dependent on type 6 secretion (T6SS) which facilitates phagosome escape. The FT T6 system is encoded on the *Francisella* pathogenicity island (FPI). However, the FT T6SS is divergent from canonical T6 systems and thus the function of many of FPI genes in T6SS remain unknown. This manuscript used a IglA-sfGFP fusion to interrogate the function of a number of FPI genes. IglA, which forms the T6SS sheath, was used to monitor T6SS dynamics (i.e. assembly, contraction and disassembly) in a variety of strains lacking genes FPI genes or the non-FPI gene ClpB. The results of these analyses led to the conclusions that IglF, IglG, IglI and IglJ were required for T6SS assemble, whereas PdpC, PdpD, PdpE, and AnmK were dispensable. Follow-up studies on the latter four genes in cultured cells and mice led to the conclusion that PdpC and PdpD were required for phagosome escape and perhaps represent secreted effectors. The manuscript is straight forward and fairly well written, but the breath of the introduction and discussion can be expanded (see below). Overall the conclusions are supported by the data. The results are an important contribution to the field and impact our understanding of the T6SS.

Major points.

1. The major shortcoming of the manuscript is that none of the described mutants have been validated. In the absence of validation it is impossible to draw comprehensive conclusions regarding the function of specific genes in the observed phenotypes. In-frame deletions have been constructed in the target genes, many of which are clustered. The only confirmation for the mutants is verifying the deletion by PCR using flanking primers. Thus, the introduction of unknown spontaneous mutations or polar effects on adjacent genes cannot be excluded.

2. There is a comprehensive analysis of many of the T6SS target genes in FT pathogenesis. However, the presented data does not delineate potential differences between pdpD and amnK. Limited data regarding a pdpD and pdpD-amnK mutant are presented, from which the authors conclude that amnK is neutral. While this may be true, the individual mutant data should be presented in the other analyses presented in Figure 5. This is particularly important as the function of pdpD in phagosomal escape is one of the major conclusions of the paper.

3. The discussion misses the opportunity to address the results in the context of the published literature. For example, the recent characterization of pdpC is not addressed in the manuscript (Microbes and Infection 18 (2016) 768e776).

Minor points.

1. Some background on the FT genes and their function in T6SS should be incorporated into the introduction. There is a nice job describing the canonical T6SS in the intro, but then the results section proceeds to assess the function of FT-specific T6SS components without a point of reference of the function of the FT genes for the uninitiated. Referencing the FT comparators to the canonical system in the intro will make the remainder of the manuscript more accessible for readers that are not well versed in T6SS.

2. Line 214 references the wrong figure.

3. The fact that ClpB is required for stress response, in addition to T6SS, indicates that its role in pathogenesis is likely pleiotropic. The presented data cannot discriminate between these two phenotypes (although the loss of T6 certainly appears to be fatal).

Reviewer #3 (Remarks to the Author):

This is a nice study that confirms that the FPI of *Francisella tularensis* encodes a functional T6SS that contracts and disassembles similar to other characterized T6SS. The authors tagged the IgIA protein with sfGFP and visualized FT T6SS assembly/disassembly both in vitro as well as within macrophages. They made some interesting observations about this unusual T6SS: they identified ClpB as the ATPase responsible for disassembly of the complex, they determined that the complex assembles preferentially at the cell poles, and they identified several of the FPI proteins as being important for complex assembly. The experiments were performed in a thorough and competent manner. The results are important and add to a thorough understanding of the basis of *Francisella* pathogenesis. Comments:

1. There have been several previous studies looking at PdpC and PdpD, and PdpC and PdpD were specifically identified as secreted effector proteins in Eshragi et al., who also showed that pdpC (and to some extent pdpD) mutants were impaired for phagosome escape. Even prior to this, Ludu et al showed that PdpD localization to the cell surface required T6SS homologues. The phagosome escape phenotype of pdpC was also shown previously by Ozanic et al, and Lindgren et al, and the virulence phenotype of pdpC was previously shown by Uda et al. So the authors of this paper

indicating that their study “suggests that PdpC and PdpD are T6SS effectors involved in phagosome rupture” is a little disingenuous, more accurate would be that their study “confirms previous studies that indicated that PdpC and PdpD are T6SS effectors...”, and citing the prior studies and findings in the discussion is important.

The *pdpD* gene is a distinguishing marker between different *Ft* species/subspecies; it is lacking in *holarctica* strains, and contains an insertion of 48 aa in *novicida* (compared to *tularensis*). Since the authors identified this effector as important for virulence, some discussion on PdpD in the various species/subspecies is warranted, since these subspecies exhibit differences in virulence.

Karl Klose

Rebuttal letter

We would like to thank all the reviewers for evaluating our manuscript. In the revised version, we addressed the critical issues raised by the reviewers. We show that *amnK* does not contribute to *F. novicida* pathogenesis. We also properly discuss the previous publications, which were pointed out by the reviewers. Point by point answers to the individual issues are below highlighted in blue.

Reviewer #1 (Remarks to the Author):

Comments:

The manuscript by Brodmann et al is focused on the T6SS dynamics and assembly and its role in anti-eukaryotic virulence in *Francisella novicida*. The authors did a very thorough dissection on the contribution of a number of previously unknown T6SS genes to the assembly, localization, and functions of T6SS using fluorescence microscopy, super-resolution microscopy, cell culture, and animal models. Novel findings include that the *Francisella* T6SS preferentially assembles on the cell pole, requires ClpB for its dynamics, and delivers putative effectors PdpC and PdpD to escape the phagosomes.

We thank the reviewer for the nice comments.

Minor comments:

1. It could be useful to include a schematic of the non-cannonical T6SS gene cluster in *Francisella*, and list corresponding known T6SS homologs with conventional nomenclature.

The scheme is now in Figure 1.

2. Figure 3 legend – line 727, should be (A) and not (B) referring to the arrow.

We have fixed the legend.

3. Should include paper by Barrigan et al 2013 “Infection with *Francisella tularensis* LVS clpB Leads to an Altered yet Protective Immune Response” as a reference when discussing the importance of ClpB for *F. novicida* pathogenesis.

This paper is now mentioned and referenced in the discussion of ClpB function (line 290-291, 297-302).

4. Line 268. “confirmed that PdpC and PdpD are potential...” here confirm and potential seem to be contradictory. Current data suggest PdpC and PdpD are effectors. However, can the authors comment on the possibility that PdpC and PdpD may not be effectors per se but rather as accessory proteins required for delivery of unidentified effectors?

We agree with the reviewer and we now discuss the possibility that PdpC and PdpD are accessory proteins necessary for the activity of yet unidentified effectors in the discussion (line 329-333).

5. This is a very minor point but it might confuse readers a little by starting the manuscript in the summary and the text with a different *Francisella* species that is not investigated in the paper. However, I will leave this to the authors judgement.

We now clearly point out this in the introduction and distinguish between *F. tularensis* and *F. novicida* throughout the text.

Reviewer #2 (Remarks to the Author):

Francisella pathogenesis is dependent on type 6 secretion (T6SS) which facilitates phagosome escape. The FT T6 system is encoded on the *Francisella* pathogenicity island (FPI). However, the FT T6SS is divergent from canonical T6 systems and thus the function of many of FPI genes in T6SS remain unknown. This manuscript used a IgIA-sfGFP fusion to interrogate the function of a number of FPI genes. IgIA, which forms the T6SS sheath, was used to monitor T6SS dynamics (i.e. assembly, contraction and

disassembly) in a variety of strains lacking genes FPI genes or the non-FPI gene ClpB. The results of these analyses led to the conclusions that IglF, IglG, IglI and IglJ were required for T6SS assemble, whereas PdpC, PdpD, PdpE, and AnmK were dispensable. Follow-up studies on the latter four genes in cultured cells and mice led to the conclusion that PdpC and PdpD were required for phagosome escape and perhaps represent secreted effectors. The manuscript is straight forward and fairly well written, but the breath of the introduction and discussion can be expanded (see below). Overall the conclusions are supported by the data. The results are an important contribution to the field and impact our understanding of the T6SS.

We thank the reviewer for these comments. As suggested by the reviewer, we have expanded the introduction and discussion to appropriately reference previous work.

Major points.

1. The major shortcoming of the manuscript is that none of the described mutants have been validated. In the absence of validation it is impossible to draw comprehensive conclusions regarding the function of specific genes in the observed phenotypes. In-frame deletions have been constructed in the target genes, many of which are clustered. The only confirmation for the mutants is verifying the deletion by PCR using flanking primers. Thus, the introduction of unknown spontaneous mutations or polar effects on adjacent genes cannot be excluded.

We have now fully sequenced all the regions, which were replaced during generation of in-frame deletions. No mutations were detected and all in-frame deletions were properly located on the chromosome. We do not show the data, however, we mention this in the Material and Methods section.

To minimize the possibility that the observed phenotypes were due to an unknown spontaneous mutation outside of the sequenced region, we checked multiple colonies obtained during generation of the in-frame deletions for T6SS sheath dynamics. In every occasion, we observed consistent results showing the same dynamics, or lack of sheath assembly, in multiple colonies with the same in-frame deletion. One of the tested colonies was then picked for further experiments.

The reviewer is right that an in-frame deletion may cause a polar effect on the downstream gene and thus complementation of in-frame deletions would strengthen our conclusions. Possible polar effect indeed complicates interpretation of a phenotype of a gene, which has the identical phenotype as deletion of the gene downstream. The rational being that if there is a polar effect then this will affect expression of the downstream gene and that could be responsible for the observed phenotype instead

of the in-frame deletion. In our set of in-frame deletions, this is a potential problem for *iglF* and *iglI*, which have identical phenotypes as the in-frame deletions of the downstream genes *iglG* and *iglJ*.

Deletion of both *iglI* or *iglJ* abolished T6SS assembly. Since we show that gene downstream of *iglJ* (*pdpC*) can be deleted without influencing T6SS assembly, we can be certain that the observed phenotype of *iglJ* is due to its deletion and not due to a polar effect on the downstream gene. However, we acknowledge that we are unable to definitively state that *iglI* is required for T6SS assembly, because the observed phenotype may be due to the polar effect on *iglJ*. In case of *iglF* and *iglG* genes, the situation is similar. However, the *iglG* was independently shown to be involved in T6SS function (Rigard et al., 2016) and thus we believe that our conclusion that both genes are required for T6SS dynamics is correct.

For all the other deletion mutants (*pdpC*, *pdpE*, *pdpD* and *amnK*), we show that the downstream genes have a different phenotype or were previously shown to have no role in FPI function.

[Redacted. The authors attempted to complement the mutants but were unsuccessful, due to apparent instability of plasmids in *F. novicida*.]

Importantly, we have noticed that in several previous *F. novicida* studies complementation from plasmid is missing (Eshraghi et al., 2016; Santic et al., 2011; Nano and Schmerk, 2007). Many times the deleted genes are reintroduced only to the same location on the chromosome (de Bruin et al., 2007; Lindgren et al., 2013; Weiss et al., 2007). This however does not eliminate the problem of a possible polar effect of an in-frame deletion, only reduces the chance that the phenotype was due to a random mutation outside of the replaced region.

Because of these technical difficulties, we modified the text to discuss the possibility of polar effects (lines 198-201), however, it has become clear that addressing this issue experimentally would unduly prolong publication of the interesting observations in our study.

2. There is a comprehensive analysis of many of the T6SS target genes in FT pathogenesis. However, the presented data does not delineate potential differences between *pdpD* and *amnK*. Limited data regarding a *pdpD* and *pdpD-amnK* mutant are presented, from which the authors conclude that *amnK* is

neutral. While this may be true, the individual mutant data should be presented in the other analyses presented in Figure 5. This is particularly important as the function of *pdpD* in phagosomal escape is one of the major conclusions of the paper.

We have now included phenotypes of *pdpD* and *amnK* single deletions. The data can be found in revised Figure 6, Supplementary Fig. S4 and Movie S1. We conclude that deletion of *amnK* gene has no measurable phenotype, neither on T6SS dynamics, vacuolar escape nor the activation of innate immunity. The results are discussed in lines 202-213, 225-232, 246-250.

3. The discussion misses the opportunity to address the results in the context of the published literature. For example, the recent characterization of *pdpC* is not addressed in the manuscript (Microbes and Infection 18 (2016) 768e776).

We have modified discussion and introduction to include this as well as other publications.

Minor points.

1. Some background on the FT genes and their function in T6SS should be incorporated into the introduction. There is a nice job describing the canonical T6SS in the intro, but then the results section proceeds to assess the function of FT-specific T6SS components without a point of reference of the function of the FT genes for the uninitiated. Referencing the FT comparators to the canonical system in the intro will make the remainder of the manuscript more accessible for readers that are not well versed in T6SS.

This is now fixed and additionally we provide a scheme in Figure 1, which labels all the genes in the FPI and provides nomenclature for canonical T6SS homologs as well as *F. novicida* gene names.

2. Line 214 references the wrong figure.

Corrected.

3. The fact that ClpB is required for stress response, in addition to T6SS, indicates that its role in pathogenesis is likely pleiotropic. The presented data cannot discriminate between these two phenotypes (although the loss of T6 certainly appears to be fatal).

This possibility is now discussed in the text (line 297-302).

Reviewer #3 (Remarks to the Author):

This is a nice study that confirms that the FPI of *Francisella tularensis* encodes a functional T6SS that contracts and disassembles similar to other characterized T6SS. The authors tagged the IglA protein with sfGFP and visualized FT T6SS assembly/disassembly both in vitro as well as within macrophages. They made some interesting observations about this unusual T6SS: they identified ClpB as the ATPase responsible for disassembly of the complex, they determined that the complex assembles preferentially at the cell poles, and they identified several of the FPI proteins as being important for complex assembly. The experiments were performed in a thorough and competent manner. The results are important and add to a thorough understanding of the basis of *Francisella* pathogenesis. Comments:

We would like to thank the reviewer for the nice comments.

1. There have been several previous studies looking at PdpC and PdpD, and PdpC and PdpD were specifically identified as secreted effector proteins in Eshragi et al., who also showed that pdpC (and to some extent pdpD) mutants were impaired for phagosome escape. Even prior to this, Ludu et al showed that PdpD localization to the cell surface required T6SS homologues. The phagosome escape phenotype of pdpC was also shown previously by Ozanic et al, and Lindgren et al, and the virulence phenotype of pdpC was previously shown by Uda et al. So the authors of this paper indicating that their study “suggests that PdpC and PdpD are T6SS effectors involved in phagosome rupture” is a little disingenuous, more accurate would be that their study “confirms previous studies that indicated that PdpC and PdpD are T6SS effectors...”, and citing the prior studies and findings in the discussion is important.

The pdpD gene is a distinguishing marker between different *Ft* species/subspecies; it is lacking in holarctica strains, and contains an insertion of 48 aa in novicida (compared to tularensis). Since the authors identified this effector as important for virulence, some discussion on PdpD in the various species/subspecies is warranted, since these subspecies exhibit differences in virulence.

We thank the reviewer for pointing out these facts. We apologize for not commenting in a sufficient detail on the previous publications. We discuss all those studies in the revised version of the manuscript.

REVIEWERS' COMMENTS:

Reviewer #2 (Remarks to the Author):

The reviewer's have satisfactorily addressed all of my comments. I have no further concerns. The revised manuscript represents an important contribution to the field and will have a broad impact our understanding of T6SS.

Reviewer #3 (Remarks to the Author):

The authors were very responsive to the comments of the reviewers and modified their manuscript accordingly. I am satisfied with the revised version of this manuscript and believe that it represents an important advance in understanding T6SS and Francisella virulence.